# Pan-Genomes Provide Insights into the Genetic Basis of *Auricularia heimuer* Domestication

**DOI:** 10.3390/jof8060581

**Published:** 2022-05-29

**Authors:** Yuxiu Guo, Zhenhua Liu, Yongping Fu, Yu Li, Yueting Dai, Shijun Xiao

**Affiliations:** 1Life Science College, Northeast Normal University, Changchun 130024, China; guoyuxiujob@163.com; 2Engineering Research Center, Chinese Ministry of Education for Edible and Medicinal Fungi, Jilin Agricultural University, Changchun 130118, China; husthuage@163.com (Z.L.); fuyongping@jlau.edu.cn (Y.F.); yuli966@126.com (Y.L.)

**Keywords:** artificial selection, chitinase, dispensable gene, glycoside hydrolase 18 family, wood ear

## Abstract

In order to reveal the genetic variation signals of *Auricularia heimuer* that have occurred during their domestication and to find potential functional gene families, we constructed a monokaryotic pan-genome of *A. heimuer* representing four cultivated strains and four wild strains. The pan-genome contained 14,089 gene families, of which 67.56% were core gene families and 31.88% were dispensable gene families. We screened substrate utilization-related genes such as the chitinase gene *ahchi1* of the glycoside hydrolase (GH) 18 family and a carbohydrate-binding module (CBM)-related gene from the dispensable families of cultivated populations. The genomic difference in the *ahchi1* gene between the wild and cultivated genomes was caused by a 33 kb presence/absence variation (PAV). The detection rate of the *ahchi1* gene was 93.75% in the cultivated population, significantly higher than that in the wild population (17.39%), indicating that it has been selected in cultivated strains. Principal component analysis (PCA) of the polymorphic markers in fragments near the *ahchi1* gene was enriched in cultivated strains, and this was caused by multiple independent instances of artificial selection. We revealed for the first time the genetic basis of the *ahchi1* gene in domestication, thereby providing a foundation for elucidating the potential function of the *ahchi1* gene in the breeding of *A. heimuer*.

## 1. Introduction

Due to genetic variation between individuals of the same species, the use of a single reference genome implies the missing of a large amount of important genetic information. In addition, basidiomycetes are dikaryotic and have two sets of diverse genomes [1]. At present, only one set of the monokaryotic genome sequence is usually used to represent all of the genetic information of the species, causing further loss of genetic information. Pan-genome technology can be used to discover more abundant variation signals during the differentiation of species and is an important approach for genomic research and breeding [2,3,4,5,6,7,8]. In particular, dispensable genes are often closely related to the establishment and environmental adaptation of important agronomic traits [9]. Therefore, the pan-genome sequencing of germplasm resources with different characteristics, such as wild and cultivated traits, may help reveal the missing genetic diversity during domestication and provide guidance for breeding [10]. Evidence of species pan-genome viability has also been confirmed in fungi [11,12,13,14,15]. However, to date, in the pan-genome study of macrofungi, there is only a single report concerning core gene families in *Lentinula edodes* [1].

In China, wild *Auricularia heimuer* is widely distributed in many regions such as Northeast China, Central China, and Northwest China [16], and the strains are adapted to complex and diverse climates and a variety of substrate conditions. The earliest history of the domestication of *A. heimuer* can be traced to before 1400 AD [17]. In the long history of domestication and breeding, strains showing elite traits such as increased yield, enhanced disease resistance, and shortened growth period were artificially selected under intensive cultivation conditions. As cultivated strains experience a series of bottleneck effects, they may have experienced greater genetic variation compared with their wild ancestors.

The first whole-genome sequence of *A. heimuer* was obtained by Yuan et al. (2017) using PacBio RSII and Illumina HiSeq 4000 sequencing platforms; the size of the genome was 49.76 Mb [18]. Dai et al. (2019) reported the genetic information of 12 germplasm resources based on whole-genome resequencing [19]. To date, the mechanism of domestication and environmental adaptation as well as the evolutionary dynamics of *A. heimuer* populations have remained unclear. In addition, a number of important functional genes are missing in the single reference genome, as well as substantial variation information. Therefore, de novo construction of the pan-genome of *A. heimuer* is necessary to obtain more comprehensive information concerning genetic variation signals in this species.

In order to fully reveal the genetic variation signals between wild and cultivated germplasm resources of *A. heimuer* and to discover important genetic variation and potential key functional gene families: (1) a monokaryotic vast pan-genome of the representative strains of *A. heimuer* was constructed, and the structural characteristics of the genome of this species were clarified; (2) the genetic basis for the artificial domestication of *A. heimuer* was revealed based on comparative genomic analysis of wild and cultivated populations; and (3) the genetic diversity of *A. heimuer* populations in Northeast China, Central China, and Northwest China were evaluated at the genome-wide level. This study provides a foundation for the effective utilization of *A. heimuer* germplasm resources and the genetic basis of domestication, and also provides a basis for guiding the breeding of *A. heimuer*.

## 2. Materials and Methods

### 2.1. Fungal Strains

We collected 48 cultivated strains and 23 wild strains of *A. heimuer* from areas in Northeast China, Central China, and Northwest China (Appendix A). Three wild strains and three cultivated strains representing the regions in Northeast China and Northwest China were selected to obtain monokaryons. The method of protoplast mononucleation reported by Dai et al. (2017) [20] was used to treat six original dikaryotic strains (the lywallzyme incubation time was regulated to 4.5 h), and finally, seven protoplast-derived monokaryons (AHD01–AHD07) were used in whole-genome sequencing, of which AHD04 and AHD05 represented the two different mating types of CCMJ1252.

### 2.2. Whole-Genome De Novo Sequencing

Genomic DNA was extracted using a NuClean Plant Genomic DNA Kit (CWBIO, Beijing, China). The concentration and purity of genomic DNA were detected using a Qubit 3.0 Fluorometer (Life Technologies, Carlsbad, CA, USA) and a Nanodrop 2000 spectrophotometer (Thermo Fisher Scientific Inc., Waltham, MA, USA), respectively. Libraries with 20 kb and 350 bp inserts were constructed, and the sequencing was performed on two long-read sequencing platforms, PacBio Sequel (Pullman, WA, USA) and Oxford Nanopore (Oxford, UK), and Illumina NovaSeq (San Diego, CA, USA), respectively, at the Novogene Bioinformatics Technology Co., Ltd. (Tianjin, China) to obtain the whole-genome sequences of the monokaryons [21,22].

### 2.3. Transcriptome Sequencing

In order to assist in accurate genome annotation, we performed transcriptome sequencing using the mature fruiting bodies of *A. heimuer*. Total RNA was extracted with TRIzol (Takara, Otsu, Japan), and the RNA quality was detected using a 2100 Bioanalyzer (Agilent Technologies, Santa Clara, CA, USA) and a NanoDrop 2000 (Thermo Fisher Scientific Inc., Waltham, MA, USA). cDNA library construction and paired-end sequencing were performed with three technical replicates performed per sample using the Illumina HiSeq X Ten platform (San Diego, CA, USA) at the Tianjin Novogene Bioinformatics Technology Co., Ltd. (Tianjin, China).

### 2.4. Whole-Genome Assembly and Annotation

We performed de novo assembly of the sequences of seven strains using NextDenovo (https://github.com/Nextomics/NextDenovo, accessed on 13 August 2021) to construct a complete whole-genome sequence of *A. heimuer*. In the same way, the published raw data of strain Dai13782 were reassembled. Racon software was used to correct the errors in the initial assembly of the third-generation sequencing data, and the result was then combined with the second-generation sequencing data. After re-correction using Pilon software, a high-quality consensus sequence was obtained. The quality of the genome assembly was assessed using Benchmarking Universal Single-Copy Orthologs (BUSCO, dataset: basidiomycota_odb9) [23,24]. Based on the Illumina data, Burrows-Wheeler Aligner (BWA) software (http://bio-bwa.sourceforge.net/, accessed on 25 August 2021) was used to evaluate the integrity of the assembly and the uniformity of sequencing.

The repetitive sequences and non-coding RNAs were annotated according to a previously reported method [19]. The coding genes were predicted based on three approaches: de novo prediction [25,26], homologous annotation, and transcriptome annotation. The protein sequences of six species (*Postia placenta* [27], *Coprinopsis cinerea* [28], *Trametes versicolor* [29], *Dichomitus squalens* [29], *A. subglabra* [29], and *Tremella mesenterica* [29]) were used as references for homologous annotation. Through BLAST searches, the predicted gene sets were compared with the data in the protein databases Swiss-Prot [30], Eukaryotic Clusters of Orthologous Groups (KOG) [31], Kyoto Encyclopedia of Genes and Genomes (KEGG, http://www.kegg.jp//, accessed on 4 October 2021), and Gene Ontology (GO) [32] to annotate the functions of those coding genes in *A. heimuer*.

### 2.5. Pan-Genome Construction

The *A. heimuer* pan-genome consisted of the genomes of seven *A. heimuer* strains sequenced in this study and the re-annotated Dai13782 reference genome. The similarity between the proteins of all species was assessed through blastp alignment, and the E value was set to 1e-5. OrthoMCL software [33] was used to find gene families via cluster analysis of the aligned data. The alignment analysis of the sequences of single-copy genes was performed using Muscle software, and then a phylogenetic tree was constructed using RAxML software. The change in the size of gene families was analyzed using Café 3.1 software [34]. The GO and KEGG enrichment analyses of the genes in expanded and contracted gene families were performed using BLAST2GO [35] and DAVID [36], respectively. Based on the frequency of a gene family present in different genomes, the genes in the *A. heimuer* pan-genome were defined as pan-gene clusters (the set of all genes); core-gene clusters (gene families present in seven or more genomes); dispensable gene clusters (gene families present in 2–6 genomes), and specific gene clusters (gene families or single genes present in only one genome).

### 2.6. Identification and Analysis of the Gene in Glycoside Hydrolase Family 18 (GH18)

MCScan software was used to analyze the regions flanking the *ahchi1* gene in eight *A. heimuer* genomes based on protein coding sequences [37]. A library with 350 bp inserts was constructed, and the whole-genome resequencing of 71 strains of *A. heimuer* was performed on the Illumina Novaseq platform (San Diego, CA, USA) at the Novogene Bioinformatics Technology Co., Ltd. (Tianjin, China). Principal component analysis (PCA) of the marker diversity in fragments near the *ahchi1* gene of the re-sequenced strains was performed using the Smartpcapl program to reveal the kinship between these strains [38]. The identification of members of the *GH18* gene family in eight genomes and the construction of the phylogenetic tree were conducted according to the methods described in Yang et al. (2021) [39]. MEME Suite 5.4.1 (http://meme-suite.org/tools/meme, accessed on 8 February 2022) was used to predict motifs. In order to verify the *ahchi1* gene, we designed specific primers (*ahchi1*F: 5′–ATGCTGCCGTTACAGTGCGG–3′ and *ahchi1*R: 5′–TCACATGTAATCATCTTGGT–3′) and used them to amplify the gene in all 71 strains of *A. heimuer*.

### 2.7. Analysis of the Genetic Diversity of A. heimuer Populations

The whole-genome sequence of AHD04 was used as a reference in the alignment analysis of resequencing data. The detection and annotation of single nucleotide polymorphisms (SNPs) followed previously reported procedures [40]. In order to clarify the evolutionary relationship between different strains of *A. heimuer*, we used four *A. cornea* species as outgroups as well as SNP data to construct a phylogenetic tree using MEGAX version 10.1 software [41] via the neighbor-joining (NJ) method.

## 3. Results

### 3.1. Whole-Genome Sequencing and De Novo Assembly

Based on the long-read sequencing and de novo assembly technology, we obtained the high-quality whole-genome sequences of seven strains and re-annotated the published sequences of the Dai13782 genome using the same parameters. The seven genomes we obtained were 47.27–49.49 Mb in size and consisted of 20–46 contigs with an N50 length of 2.72–3.90 Mb and contained 56.91–57.04% GC (Table 1). The rate of complete BUSCOs was 95.80–96.3%, indicating that the genomes we obtained had decent completeness. The alignment rate of short-fragment high-quality Illumina reads was 93.96–99.37%; the genome coverage rate was 93.15–99.79%, and the average coverage depth was 69.66–136.88X. These results showed that the above Illumina reads were in excellent consistency with the assembled genome. In addition, we obtained 9.55 Gb transcriptome data with a GC content of 60% for subsequent functional gene annotation.

### 3.2. Gene Prediction and Annotation

Our analysis predicted that there were 13,415–13,836 protein coding genes in the eight genomes of *A. heimuer*. The average length of protein coding genes was 2302.85–2559.63 bp. The average length of coding sequences (CDS) was 1351.37–1395.87 bp. On average, each gene contained 6.25–6.48 exons. The average exon and intron lengths were 291.23–304.81 bp and 89.6–106.86 bp, respectively (Appendix A). In addition, a 7.70–9.13 Mb fragment in the eight genomes was annotated as a repetitive sequence, accounting for 15.52%–18.34% of the full genome size. The 4.34–5.10 Mb retrotransposons accounted for 8.94%–10.24% of the whole genome, in which the 3.56–4.22 Mb long terminal repeats (LTR) were the most abundant, accounting for 7.49%–8.48% of the genome size (Appendix A). There was a highly significant correlation between the genome size and the content of repetitive sequences in the genomes. The higher the repetitive sequence count, the larger the genome size (R = 0.79, *p* = 0.0032). Moreover, 172–213 ncRNAs, including 138–145 tRNAs, 9–45 rRNAs, and 24–26 snRNAs, were also predicted in the eight *A. heimuer* genomes (Appendix A).

### 3.3. A. heimuer Pan-Genome

The 109,176 annotated genes in the eight genomes were classified into 91,973 gene families, with an average of 1.148–1.166 genes per gene family, of which AHD07 had the largest number of gene families (11,700) and AHD02 had the least number of gene families (11,314, Figure 1A). The Dai13782 genome contained the largest number of unique gene families (22) and unique genes (48). The AHD05 genome contained the least number of unique gene families (3) and unique genes (6, Figure 1B and Appendix A). Functional annotation indicated that these unique genes were related to basic life processes, such as DNA-binding domain, reverse transcriptase, and ubiquitin-protein ligase. A total of 8100 gene families involving 9459–9604 genes present in all eight genomes. Eight *A. heimuer* genomes had up to 6990 single-copy gene families.

Referring to the pan-genome definition of Brassica napus [7], a vast pan-genome of *A. heimuer* was constructed using the above-mentioned eight *A. heimuer* reference genomes, of which four genomes represented wild strains and four genomes represented cultivated strains. The *A. heimuer* pan-genome covered 14,089 gene clusters, of which 67.56% (9519) were core gene families present in at least seven genomes; 31.88% (4492) were dispensable gene families present in two to six genomes, and about 0.56% (78 gene families and 5041 single genes) were unique gene families with no orthologs in other genomes (Table 2; Figure 1B,C). Functional annotation showed that the core gene families were involved in the ABC transporter, peptidase S8 family domain, and transcriptional regulator ICP4, etc. Dispensable gene families were related to the 2OG-Fe (II) oxygenase superfamily, C2H2 Zn finger, and DNA polymerase III, etc. The unique gene families were related to reverse transcriptase, mitochondrial chaperone BCS1, and DEAD-like helicases superfamily, etc.

Using *A. cornea* as an outgroup, we constructed a phylogenetic tree based on single-copy gene families, showing that the eight *A. heimuer* strains were clearly classified into Northeast China and Northwest China groups, and the Northeast China group was further divided into wild and cultivated subgroups (Figure 1D). During the evolution of *A. heimuer*, the size of 243 gene families expanded, and 1562 gene families contracted. We noticed that the significantly expanded gene families in *A. heimuer* were involved in the chitin metabolic process, and the significantly contracted gene families were involved in serine hydrolase activity and peptidase activity. Within *A. heimuer* species, as in other macrofungi, the number of contracted gene families was much greater than that of the expanded gene families. Within the Northeast China group, 97 gene families were expanded, and 897 gene families were contracted during the evolution of cultivated strains.

### 3.4. Mining Cultivar-Specific Genes in A. heimuer

In order to discover functional genes associated with the domestication of *A. heimuer*, we analyzed the dispensable gene families that were present simultaneously and only in the genomes of the four cultivated strains AHD04 to AHD07. The numbers of this kind of gene in AHD04 to AHD07 were 46, 47, 47, and 48, respectively. Functional annotation showed that a chitinase gene *ahchi1* of the GH18 family (*DJ50020910.1*, *DJ60112240.1*, *H290116510.1*, and *T-20115100.1*) was present in the genomes of all four cultivated strains. In addition, the four cultivated genomes all contained a carbohydrate-binding module family 50 (CBM50, *DJ50020900.1*, *DJ60112230.1*, *H290116500.1*, and *T-20115090.1*) and a terpenoid synthase gene (*DJ50106660.1*, *DJ60097650.1*, *H290112180.1*, and *T-20081890.1*). Discovering dispensable genes in the cultivated populations provided an important basis for elucidating the mechanism of *A. heimuer* domestication and understanding phenotypic specificity.

We identified a total of 76 genes in the GH18 family in the eight genomes. Eight to ten genes in the GH18 family were predicted in each of the four cultivated genomes. The phylogenetic tree constructed using the protein sequences of 76 GH18 genes showed that the strains were clustered into three main groups (I, II, and III, Figure 2). Group II was further divided into four subgroups (A, B, C, and D), of which the four *ahchi1* genes constituted subgroups II-C. This result was consistent with that obtained through motif prediction.

In order to explore the origin of the *ahchi1* gene in wild and cultivated resources, we compared the distribution of genes flanking *ahchi1*. Based on the collinearity of protein-coding genes, we found that although the genomes of the wild strains analyzed in this study did not contain the *ahchi1* gene, the homologous genes flanking the *ahchi1* gene could be obtained by homologous alignment. These genes showed a very clear conserved collinearity of gene arrangement in both wild and cultivated strains (Figure 3A). For example, the genes *DJ50020880.1*/*DJ50020890.1* and *DJ50020940.1*/*DJ50020950.1* were present in both wild and cultivated strains and were collinearly distributed on two sides of the genomic segment where the *ahchi1* gene resided. In all the cultivated strains, the genome fragment containing the *ahchi1* gene formed a 33 kb presence/absence variation (PAV) compared with the wild strains, and this affected five conserved genes including the *ahchi1* gene in both wild and cultivated strains. It was the insertion of this specific genome fragment that enabled the monokaryotic genome of the cultivated strains to obtain the specific *ahchi1* gene.

In order to systematically identify the distribution of the *ahchi1* gene in wild and cultivated strains and to verify its role in the domestication of *A. heimuer*, we detected the *ahchi1* gene in all of the 71 *A. heimuer* strains. The gene was successfully amplified in 45 of 48 cultivated strains and in 4 of 23 wild strains. The detection rates of this gene in cultivated and wild populations were 93.75% and 17.39%, respectively.

The alignment of 71 whole-genome resequencing data to the reference genome AHD04 showed that the average mapping rate and coverage rate was 88.32% and 88.97%, respectively (Appendix A). A total of 1,444,490 high-quality SNP sites were screened from the 71 strains. Among these, 153,998 mutations were non-synonymous and covered 13,836 genes, of which 284,755 were located in coding regions; 187,601 were located in intron regions; 157,370 were located upstream of genes, and 168,329 were located downstream of genes. Those variations are useful for the genetic and breeding research of *A. heimuer*.

In order to further explore the effect of artificial selection on the enrichment of the *ahchi1* gene, we performed cluster analysis using the polymorphic markers flanking the *ahchi1* gene. PCA analysis showed that the distribution of the wild strains was scattered, while the distribution of cultivated strains was concentrated to some extent and was classified into two main clusters comprising 19 and 27 cultivated strains, respectively (Figure 3B). The results indicated that the enrichment of the *ahchi1* gene in cultivated strains was the result of multiple rounds of artificial selection.

### 3.5. Evaluation of the Population Genetic Diversity of A. heimuer

In order to clarify the phylogenetic relationship between the germplasm resources of *A. heimuer*, we constructed a phylogenetic tree based on SNPs and used four *A. cornea* as the outgroup (Figure 4). The wild strains collected from Northwest China were grouped in clade I. The strains from Central China were clustered into a single subgroup (clade II). The rest of the strains collected from Northeast China and the cultivated strains from Northwest China formed clade III. Clade II strains had a closer relationship with clade III strains compared with clade I. In addition, some cultivated and wild strains could not be clearly distinguished in clade III.

## 4. Discussion

The seven high-quality genomes we obtained consisted of 20–46 contigs, with contig N50 ranging between 2.72 and 3.90 Mb, the completeness of which was significantly higher than the previously reported genome that had 103 contigs and a contig N50 of 1.35 Mb [18]. Three genomes of the *Auricularia* genus have been reported. The average genome size of *A. heimuer* was 48.67 Mb, 0.65–0.66 times smaller than those of *A. cornea* (73.48 Mb) and *A. subglabra* (74.92 Mb) [19,29]. The average length of repetitive sequences in the *A. heimuer* genome was 8.32 Mb, accounting for 17.09% of the genome size, significantly smaller than the average length (15.05 Mb) and proportion (20.48%) of the repetitive sequences in the *A. cornea* genome. In addition, the average number (13,647) and length (2440 bp) of the coding genes in the *A. heimuer* genome were lower than the average number (17,591) and length (2541 bp) of the coding genes in reported *A. cornea* genomes [19]. It has been reported that the differences in repetitive sequences are the main cause of genome size variation between *Pleurotus* species [42]. Therefore, the above-mentioned differences in repetitive sequences and coding genes might lead to the diverse genomes size of the different *Auricularia* species.

The construction of a pan-genome is of great importance for mining important functional genes and guiding breeding. We constructed a vast pan-genome of *A. heimuer* including wild and cultivated strains, and this has provided a foundation for the identification of variants at the genome level and the analysis of genetic diversity and phenotypic heterogeneity. There were 87,655 coding genes in this pan-genome involving 14,089 gene families that included about 67.56% core gene families, 31.88% dispensable gene families, and 0.56% specific gene families. The content of core gene families in the pan-genome of *A. heimuer* was close to those of *Saccharomyces cerevisiae* (63.37%) [14] and *Aspergillus fumigatus* (69%) [11]. The difference in the number of core genes in different genomes may be due to the diverse quality of genome assembly, which may result in discrepancies in the types and numbers of genes predicted. Like *L. edodes* [1], the core genes of *A. heimuer* covered the genetic programs of basic life processes such as cell growth and fruiting body development. In the future, larger-scale research into the *A. heimuer* pan-genome is necessary to discover the key gene families associated with agronomic traits.

As an economically important food crop, the evolutionary dynamics related to the domestication of edible fungi are still unclear. We attempted to fill in the gaps through mining the genes potentially involved in the domestication by comparing the *A. heimuer* genomes of wild and cultivated strains. We found several cultivar-specific genes, of which the *ahchi1* gene of the GH18 gene family was cloned and identified. The results also confirmed that the genome annotation was accurate. Chitinases (EC 3.2.1.14) primarily function to catalyze the hydrolysis of the β-1,4 bond that links chitin and chitosan oligomer, resulting in the release of short-chain chitooligosaccharide [43,44,45]. The GH18 chitinase gene family has been shown to be involved in fungal morphogenesis, including spore germination, mycelia elongation, and branching [46,47]. Zhou et al. found that the chitinase ChiE1 and ChiIII in *Coprinopsis cinerea* enabled the elongation of the heat-inactivated cell walls in stipe [48]. We found that the detection rate of the *ahchi1* gene in the cultivated populations (93.75%) was significantly higher than that in the wild populations (17.39%), indicating that the *ahchi1* gene was significantly enriched in the cultivated population by artificial selection, possibly due to the fact that the *ahchi1* gene can promote the growth and development of mycelia and fruiting bodies of *A. heimuer*. Long-term domestication and breeding activities tend to retain the strains with faster development. Therefore, artificial selection to a certain extent promoted the preferential enrichment and rapid evolution of the *ahchi1* gene in cultivated populations. Furthermore, by comparing genome collinearity, we found that a 33 kb genomic fragment containing the *ahchi1* gene was specific to cultivated strains, and the protein-coding genes in the fragment flanking region had clearly conserved collinearity. Therefore, we speculate that the enrichment of the *ahchi1* gene in the genome of cultivated strains was achieved through the PAV of the 33 kb fragment. Based on the population genetic analysis of the markers flanking this segment, we furtherly found that the cultivars were clustered into two main groups, suggesting that the *ahchi1* gene in the cultivars was selected through multiple rounds of artificial selection. In this study of the *A. heimuer* pan-genome, we revealed for the first time the genetic basis of the *ahchi1* gene during domestication, thereby providing a basis for further elucidation of the potential function of the *ahchi1* gene and its utilization in breeding.

The phylogenetic tree, constructed based on whole-genome resequencing data, showed that the *A. heimuer* strains collected from Central China and Northeast China were more closely related, a result that was consistent with the geographic distance of the three ecological regions and the developmental history of *A. heimuer*. Our study provided useful data for exploring niche-specific adaptive evolution and gene exchange of *A. heimuer* strains. Having a suitable climate and wood substrates for *A. heimuer*, Northeast China has become the main producing area of *A. heimuer* in China and also has more abundant wild and cultivated resources of this species. The *A. heimuer* strains collected from Northeast China were important materials in this study. It is worth noting that the wild and cultivated strains of *A. heimuer* are indistinguishable at the level of genome-wide genetic variation, implying that some of the wild and cultivated strains of *A. heimuer* have similar genetic backgrounds. The same phenomenon also occurs in *L. edodes* [49]. Unfortunately, we do not know exactly the original background of all cultivated strains. Some of the cultivated strains may have been simply domesticated for a short period of time based on wild strains, and some other strains may be the product of cross breeding between wild and cultivated parents; these two types of domesticated strains still have a strong background from their wild ancestors. In this study, we found that some wild strains, especially the germplasm resources collected from Northwest China, contained more unique gene pools, and these wild resources are important for the improvement of existing local cultivated strains.

## Figures and Tables

**Figure 1 jof-08-00581-f001:**
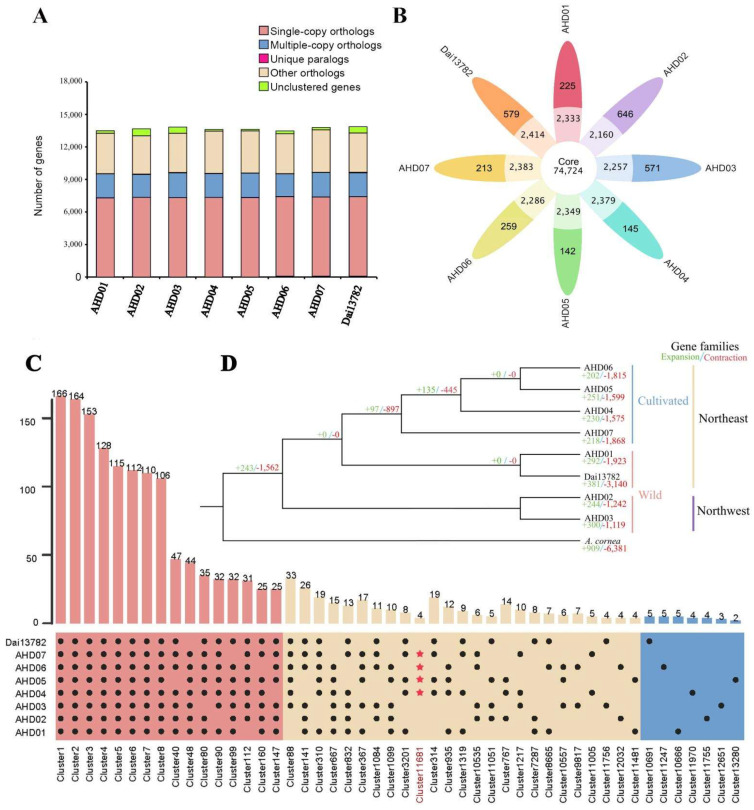
The pan-genome and phylogenetic relationship of eight *A. heimuer* accessions: (**A**) Comparison of orthologous genes; (**B**) Flower plot of core gene clusters (present in seven or eight genomes), dispensable gene clusters (present in two to six genomes), and singletons. The number of gene families is shown in each of the diagram components; (**C**) Core- and pan-genome histograms. Four red stars represent the dispensable cluster 11,681 consisting of four chitinase genes that only existed in cultivated strains; (**D**) Phylogenetic analysis of *A. heimuer* and *A. cornea*. The green and red numbers represent expanded and contracted gene families.

**Figure 2 jof-08-00581-f002:**
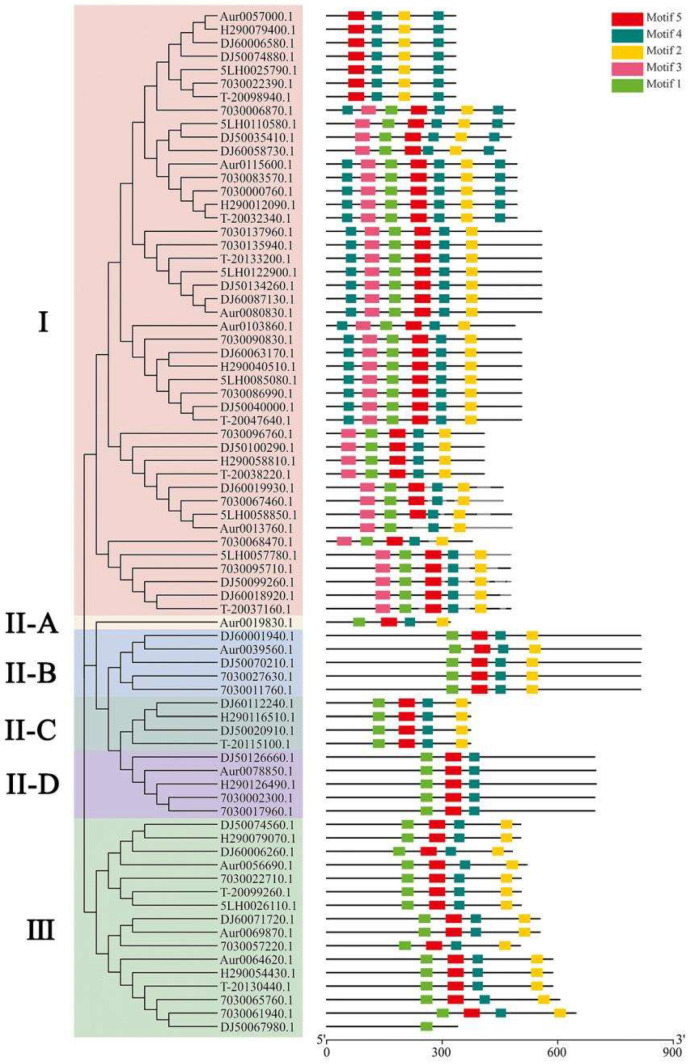
Phylogenetic relationship and motif analysis of 76 GH18 genes: (**A**) Phylogenetic tree. The 76 GH18 proteins are clustered into six groups. Group II-C consisted of four subgroups and four *ahchi1* proteins. (**B**) Motif analysis. The length and different colors of boxes denote motif length and different motifs, respectively. Group II-C was quite different from other members of Group II.

**Figure 3 jof-08-00581-f003:**
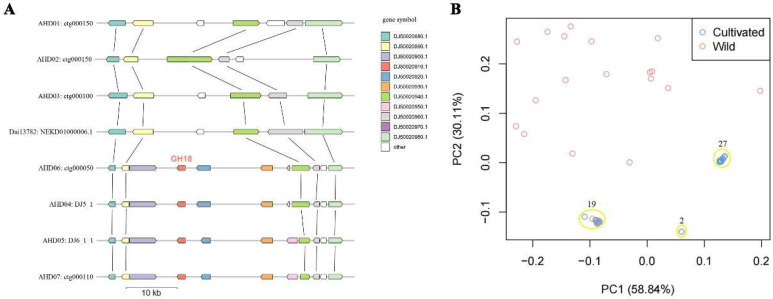
Collinearity and diversity analysis of fragments around ahchi1 gene of GH18 gene family: (**A**) The conserved synteny for neighboring genes of ahchi1 gene. AHD04 genes are used for symbols. (**B**) PCA analysis for populations using markers around the specific genomic fragments containing the ahchi1 gene.

**Figure 4 jof-08-00581-f004:**
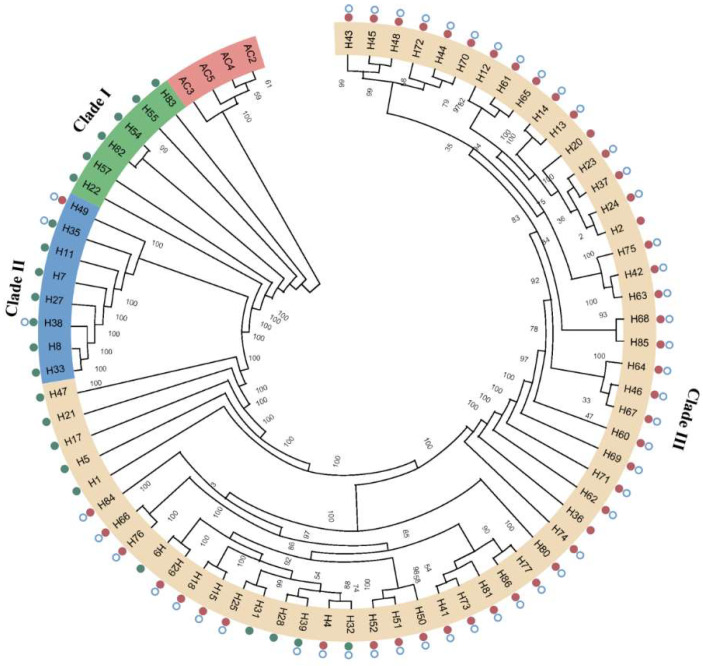
A neighbor-joining (NJ) tree of 71 *A. heimuer* accessions based on the whole-genome SNPs. All samples clustered into northwestern China (Clade I), central China (Clade II), and northeast China (Clade III) groups. Cultivated and wild strains are labeled using red and green dots, respectively. Strains that have the *ahchi1* gene are labeled with a blue hollow circle.

**Table 1 jof-08-00581-t001:** Statistics of genomic assembly and annotation for eight *A. heimuer* genomes.

Accession	AHD01	AHD02	AHD03	AHD04	AHD05	AHD06	AHD07	Dai13782
Genome size (Mb)	49.49	47.27	48.49	48.74	49.34	47.42	48.86	49.76
Coverage (X)	119	151	138	216	351	284	136	57
Number of contigs	35	20	24	26	25	27	46	103
N50 (Mb)	3.32	3.90	3.88	3.27	3.55	3.16	2.72	1.35
GC content (%)	57.02	57.04	56.97	56.94	56.93	56.91	56.97	56.94
Complete BUSCOs (%)	95.90	96.00	96.30	95.80	96.20	96.30	96.30	94.00
Repetitive elements (Mb)	8.61	7.34	7.70	8.60	8.72	7.93	8.53	9.13
LTR (Mb)	4.01	3.65	3.63	3.69	3.72	3.56	3.76	4.22
Number of genes	13,490	13,673	13,836	13,609	13,620	13,415	13,730	13,803
Average gene length (bp)	2559.63	2302.85	2306.09	2550.24	2549.24	2441.28	2399.34	2412.73
Average CDS length (bp)	1392.24	1386.41	1380.84	1386.86	1395.87	1379.86	1382.21	1351.37
Average exon per gene	6.48	6.28	6.25	6.48	6.45	6.34	6.33	6.41
Average exon length (bp)	304.67	291.23	292.85	303.16	304.81	298.52	301.55	300.84
Average intron length (bp)	106.86	89.6	90.5	106.71	106.66	102.37	92.3	89.62

**Table 2 jof-08-00581-t002:** Statistics of the *A. heimuer* pan-genome.

Accession	Core Family	Core Gene	Dispensable Family	Dispensable Gene	Specific Gene Cluster	Singletons	Total ^a^
AHD01	9327	10,916	2169	2333	5	225	11,726
AHD02	9295	10,832	2003	2160	16	646	11,960
AHD03	9302	10,971	2079	2257	18	571	11,970
AHD04	9444	11,068	2187	2379	6	145	11,780
AHD05	9437	11,123	2182	2349	3	142	11,764
AHD06	9344	10,858	2114	2286	4	259	11,721
AHD07	9458	11,125	2239	2383	4	213	11,913
Dai13782	9127	10,762	2189	2414	22	579	11,917
All	74,734	87,655	17,162	18,561	2960	2780	94,751

^a^: Total of clusters and singletons.

## Data Availability

The datasets presented in this study can be found in online repositories. The raw genome sequencing data of *Auricularia heimuer* were deposited on NCBI linked to BioProject: PRJNA812637, PRJNA813000, PRJNA812997, PRJNA812994, PRJNA812377, PRJNA812383, and PRJNA812626 and BioSample: SAMN26421298, SAMN26443526, SAMN26443398, SAMN26443385, SAMN26377383, SAMN26377460, and SAMN26421122 within GenBank. The transcriptome data were submitted to FigShare (10.6084/m9.figshare.19888816). Data generated or analyzed during this study are included in this published article and its Appendix A.

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
