# Peer review of "Pan-Genomes Provide Insights into the Genetic Basis of Auricularia heimuer Domestication"

_jof, 2022, doi:10.3390/jof8060581_

Round 1

Reviewer 1 Report

Pangenomic sequencing means the analysis of a set of genes of a certain population, species or taxon of a higher rank, within which the exchange of genetic information is possible. When working with a single strain, a large amount of important genetic information remains unknown. In the case of fungi, the dikaryotic nature of the genome should also be taken into account. The desire for a more complete identification of genes allows expanding the possibilities of selection. The object of this work is the basidiomycete Auricularia heimuer, which has been cultivated in China for hundreds of years. The natural diversity of A. heimuer populations is explained by its wide distribution in China and existence in different natural conditions.

The subjects were two groups of A. heimuer strains: 48 cultivated and 23 wild. From these groups, 3 strains were selected for whole genome sequencing. 7 monokaryons were obtained. On the basis of genome-wide sequences, as well as taking into account the data on the reference strain, a pangenomic pattern of A. heimuer was created. Using 8 strains, the ahchi1 gene of the Glycoside Hydrolase Family 18 (GH18) was identified and studied. For this, a library was constructed and whole genome re-sequencing of 71 strains of A. heimuer was carried out, the degree of relationship was determined.

The experiments were performed on modern equipment using a large number of molecular biology methods. The work is fundamental, a huge amount of materials is presented. It has both applied and fundamental significance.

With such a volume of material, inevitable remarks can be noted, which, nevertheless, are controversial. For example, the expression "We constructed a comprehensive pan-genome of A. heimuer ...." too brave, which I would replace with "We constructed a vast pan-genome of A. heimuer ....". However, I do not insist on changing the wording.

The following typos should be corrected:

line 82: “dinokaryotic strains” This term is used for dinoflagellates. How appropriate is its use in relation to fungus?

line 251: “lusters and singeletons” Replace with “Clusters and singeletons”

The work is good and can be published with minimal editing.

Author Response

Thank you for your comments. Our answers to your points are as follows.

  1. With such a volume of material, inevitable remarks can be noted, which, nevertheless, are controversial. For example, the expression "We constructed a comprehensive pan-genome of A. heimuer ...." too brave, which I would replace with "We constructed a vast pan-genome of A. heimuer ....". However, I do not insist on changing the wording.

         Response 1.

         Thank you. We agree with the reviewer and have revised “comprehensive”  to “vast” of the manuscript.

         Line 67. “a monokaryotic vast pan-genome…”.

         Line 208 and 349. “a vast pan-genome of A. heimuer…”.

  1. The following typos should be corrected:

          Line 82: “dinokaryotic strains” This term is used for dinoflagellates. How appropriate is its use in relation to fungus?

         Response 2.

         Sorry for this mistake. We have revised “dinokaryotic strains” to “dikaryotic strains”.

  1. line 251: “lusters and singeletons” Replace with “Clusters and singeletons”

         Response 3.

         Sorry for that. We have revised the footnote to “a: Total of clusters and singeletons”.

Reviewer 2 Report

The manuscript report on a great work of genome resequencing and data analyses using the modern tools for identifying genetic basis correlated with the domestication of an edible mushroom by comparing cultivated and wild strains.

Following are few specific comments:

Line 22: ‘PCA analysis’ of what? the whole-genome re-sequencing of 71 strains from a library with 350 bp inserts in flanking regions of ahchi1 gene? Information should be given in abstract.

Line 96: I did not find any data on transcriptome sequencing in Results or supplementary materials. You should delete the paragraph or add in Results how the transcriptome was used and where data are available.

Line 99 : which experiment was replicated? Does it mean 3 different fruiting bodies?

Line 251: be careful, the footnote moved partly vertically

Line 281: PAV is defined in abstract, but it should be also defined here: presence/absence variation

Line 293: Group VI, maybe you referred to group II.

Line 308: Change mian to main.

Line 353: 67.56% could be deleted here as it is in the previous sentence.

Author Response

Thank you for your comments. Our answers to your points are as follows.

  1. Line 22: ’PCA analysis’ of what? the whole-genome re-sequencing of 71 strains from a library with 350 bp inserts in flanking regions of ahchi1 gene? Information should be given in abstract.

          Response 1.

          Thank you. The “PCA analysis” were performed using the polymorphic markers in fragments near the ahchi1 gene which were located from 1145900–1162700 bp in reference genome AHD04. As suggested, we have added more details in abstract line 22 “Principal component analysis (PCA) of the polymorphic markers in fragments near the ahchi1 gene”.

  1. Line 96: I did not find any data on transcriptome sequencing in Results or supplementary materials. You should delete the paragraph or add in Results how the transcriptome was used and where data are available.

          Response 2.

          We are sorry for that. We have added more details in Results line 179–180 and 187,  Supplementary materials line 417–422, and Data Availability Statement line 445–446.

           Line 179–180. “In addition, we obtained 9.55 Gb transcriptome data with a GC content of 60% for subsequent functional gene annotation.”

            Line 187. “(Table S2–S9)”.

            Line 417–422. “Table S2: Prediction of the coding genes in AHD01 genome, Table S3: Prediction of the coding genes in AHD02 genome, Table S4: Prediction of the coding genes in AHD03 genome, Table S5: Prediction of the coding genes in AHD04 genome, Table S6: Prediction of the coding genes in AHD05 genome, Table S7: Prediction of the coding genes in AHD06 genome, Table S8: Prediction of the coding genes in AHD07 genome, Table S9: Prediction of the coding genes in Dai13782 genome”.

            Line 445–446. “The transcriptome data was submitted to FigShare (10.6084/m9.figshare.19888816)”.

Table S2. Prediction of the coding genes in AHD01 genome.

Gene set

Number

Average gene length (bp)

Average CDS length (bp)

Average exon per gene

Average exon length (bp)

Average intron length (bp)

denovo/GlimmmerHMM

15,572

1,745.51

1,290.26

4.24

303.97

140.31

denovo/AUGUSTUS

16,480

1,793.90

1,432.09

5.91

242.29

73.68

homo/Trametes_versicolor

12,564

6,004.41

805.39

3.86

208.38

1,814.71

homo/Dichomitus_squalens

12,005

7,535.14

820.46

3.99

205.81

2,248.34

homo/Auricularia_subglabra

22,841

7,913.57

1,006.45

4.11

244.69

2,218.67

homo/Coprinus_cinereus

11,200

6,666.13

806.77

3.90

207.10

2,023.52

homo/Tremella_mesenterica

6,422

5,943.71

767.49

3.69

207.81

1,921.93

homo/Postia_placenta

11,076

5,661.53

787.94

3.94

199.83

1,655.98

trans.orf/RNAseq

5,761

3,429.84

1,582.81

8.13

333.08

101.01

BUSCO

1,329

2,981.23

1,935.98

9.21

210.31

127.39

MAKER

11,478

2,883.46

1,439.17

6.93

313.35

120.19

HiCESAP

13,490

2,559.63

1,392.24

6.48

304.67

106.86

Table S3. Prediction of the coding genes in AHD02 genome

Gene set

Number

Average gene length (bp)

Average CDS length (bp)

Average exon per gene

Average exon length (bp)

Average intron length (bp)

denovo/GlimmmerHMM

15,940

1,658.76

1,263.92

4.20

301.28

123.58

denovo/AUGUSTUS

16,183

1,810.56

1,449.24

5.95

243.60

73.00

homo/Coprinus_cinereus

10,497

6,284.57

814.68

3.95

206.37

1,855.71

homo/Tremella_mesenterica

6,021

4,752.87

770.88

3.72

207.09

1,462.64

homo/Postia_placenta

10,548

5,927.19

798.16

3.99

200.20

1,717.22

homo/Dichomitus_squalens

11,220

6,691.86

828.07

4.02

205.86

1,940.07

homo/Trametes_versicolor

11,727

5,480.29

813.08

3.91

207.78

1,602.10

homo/Auricularia_subglabra

22,108

8,604.87

1,013.45

4.15

244.43

2,412.93

trans.orf/RNAseq

5,284

3,085.49

1,540.39

7.75

316.69

93.66

BUSCO

1,334

2,934.67

1,905.55

9.09

209.62

127.20

MAKER

11,586

2,559.26

1,436.98

6.72

296.27

99.20

HiCESAP

13,673

2,302.85

1,386.41

6.28

291.23

89.60

Table S4. Prediction of the coding genes in AHD03 genome

Gene set

Number

Average gene length (bp)

Average CDS length (bp)

Average exon per gene

Average exon length (bp)

Average intron length (bp)

denovo/GlimmmerHMM

15,891

1,683.40

1,266.78

4.19

302.25

130.55

denovo/AUGUSTUS

16,438

1,803.62

1,442.32

5.96

242.08

72.87

homo/Coprinus_cinereus

10,682

5,768.38

806.25

3.88

207.96

1,724.82

homo/Postia_placenta

10,677

5,346.71

791.56

3.94

200.92

1,549.59

homo/Dichomitus_squalens

11,310

5,876.39

825.92

4.01

205.86

1,676.72

homo/Auricularia_subglabra

22,747

8,400.02

998.69

4.09

244.24

2,396.04

homo/Trametes_versicolor

12,082

5,947.66

808.38

3.89

207.94

1,779.77

homo/Tremella_mesenterica

6,117

4,621.17

774.88

3.70

209.20

1,422.48

trans.orf/RNAseq

5,475

3,072.29

1,551.48

7.79

316.38

89.26

BUSCO

1,335

2,971.85

1,934.48

9.24

209.30

125.85

MAKER

11,704

2,576.53

1,434.95

6.71

298.33

100.51

HiCESAP

13,836

2,306.09

1,380.84

6.25

292.85

90.50

Table S5. Prediction of the coding genes in AHD04 genome

Gene set

Number

Average gene length (bp)

Average CDS length (bp)

Average exon per gene

Average exon length (bp)

Average intron length (bp)

denovo/GlimmmerHMM

15,620

1,711.68

1,277.40

4.21

303.77

135.49

denovo/AUGUSTUS

16,417

1,787.75

1,432.64

5.89

243.25

72.63

homo/Tremella_mesenterica

6,216

5,490.37

780.72

3.74

208.49

1,715.91

homo/Postia_placenta

10,925

5,750.72

795.46

3.97

200.37

1,668.45

homo/Trametes_versicolor

12,282

5,639.07

811.67

3.88

209.05

1,674.63

homo/Dichomitus_squalens

11,674

5,952.55

822.47

3.99

206.23

1,716.79

homo/Coprinus_cinereus

11,007

5,779.31

809.44

3.91

207.18

1,709.69

homo/Auricularia_subglabra

22,597

8,225.36

1,009.77

4.11

245.84

2,322.06

trans.orf/RNAseq

5,925

3,495.98

1,598.08

8.16

331.09

110.96

BUSCO

1,327

2,954.08

1,916.18

9.16

209.13

127.15

MAKER

11,447

2,805.52

1,441.31

6.96

311.92

106.13

HiCESAP

13,609

2,550.24

1,386.86

6.48

303.16

106.71

Table S6. Prediction of the coding genes in AHD05 genome

Gene set

Number

Average gene length (bp)

Average CDS length (bp)

Average exon per gene

Average exon length (bp)

Average intron length (bp)

denovo/GlimmmerHMM

16,074

1,751.05

1,299.30

4.26

304.76

138.43

denovo/AUGUSTUS

16,550

1,800.38

1,441.58

5.91

244.02

73.11

homo/Tremella_mesenterica

6,254

4,812.58

775.47

3.75

206.77

1,467.83

homo/Postia_placenta

11,027

5,799.68

796.24

3.97

200.41

1,682.87

homo/Coprinus_cinereus

11,063

5,898.33

812.79

3.90

208.47

1,754.33

homo/Auricularia_subglabra

22,802

8,089.95

1,003.12

4.09

245.12

2,291.72

homo/Trametes_versicolor

12,428

5,885.78

811.23

3.88

208.89

1,759.86

homo/Dichomitus_squalens

11,695

6,276.52

820.87

3.99

205.65

1,823.64

trans.orf/RNAseq

5,941

3,509.65

1,579.31

8.03

333.57

118.04

BUSCO

1,335

2,991.73

1,925.42

9.32

206.59

128.16

MAKER

11,517

2,842.99

1,456.13

6.95

313.66

111.60

HiCESAP

13,620

2,549.24

1,395.87

6.45

304.81

106.66

Table S7. Prediction of the coding genes in AHD06 genome

Gene set

Number

Average gene length (bp)

Average CDS length (bp)

Average exon per gene

Average exon length (bp)

Average intron length (bp)

denovo/GlimmmerHMM

15,520

1,703.81

1,276.81

4.21

303.20

132.98

denovo/AUGUSTUS

16,088

1,796.99

1,439.68

5.92

243.07

72.58

homo/Tremella_mesenterica

6,213

5,127.98

771.00

3.69

209.05

1,620.86

homo/Dichomitus_squalens

11,449

6,132.30

824.30

4.01

205.79

1,766.09

homo/Auricularia_subglabra

22,417

7,954.35

1,011.88

4.10

246.94

2,241.14

homo/Trametes_versicolor

12,165

5,354.01

809.52

3.87

209.23

1,583.97

homo/Postia_placenta

10,853

5,314.61

787.16

3.93

200.27

1,544.93

homo/Coprinus_cinereus

10,927

6,036.01

805.70

3.89

207.30

1,811.92

trans.orf/RNAseq

5,742

3,279.63

1,571.18

7.91

325.63

101.95

BUSCO

1,333

2,966.48

1,923.67

9.29

207.18

125.87

MAKER

11,474

2,661.49

1,436.89

6.83

304.20

100.30

HiCESAP

13,415

2,441.28

1,379.86

6.34

298.52

102.37

Table S8. Prediction of the coding genes in AHD07 genome

Gene set

Number

Average gene length (bp)

Average CDS length (bp)

Average exon per gene

Average exon length (bp)

Average intron length (bp)

denovo/GlimmmerHMM

15,726

1,757.26

1,328.73

4.20

316.55

134.02

denovo/AUGUSTUS

16,472

1,801.29

1,437.07

5.94

242.05

73.77

homo/Postia_placenta

11,043

5,398.27

794.97

3.95

201.16

1,559.43

homo/Dichomitus_squalens

11,826

6,086.15

828.69

3.99

207.51

1,756.30

homo/Coprinus_cinereus

11,206

6,332.97

806.80

3.88

208.01

1,919.72

homo/Trametes_versicolor

12,318

5,352.83

816.18

3.89

209.80

1,569.60

homo/Auricularia_subglabra

22,962

7,794.25

1,015.94

4.10

248.06

2,189.76

homo/Tremella_mesenterica

6,416

5,044.13

776.57

3.69

210.33

1,585.17

trans.orf/RNAseq

5,592

3,199.77

1,557.30

7.96

325.88

86.87

BUSCO

1,335

2,973.51

1,927.98

9.16

210.40

128.08

MAKER

11,806

2,610.87

1,436.10

6.77

309.59

89.38

HiCESAP

13,730

2,399.34

1,382.21

6.33

301.55

92.30

Table S9. Prediction of the coding genes in Dai13782 genome

Gene set

Number

Average gene length (bp)

Average CDS length (bp)

Average exon per gene

Average exon length (bp)

Average intron length (bp)

denovo/GlimmmerHMM

16,019

1,795.56

1,355.40

4.16

325.64

139.19

denovo/AUGUSTUS

16,388

1,789.72

1,414.07

6.02

235.06

74.89

homo/Postia_placenta

11,761

4,720.55

787.93

3.93

200.40

1,341.36

homo/Tremella_mesenterica

6,893

5,019.07

778.07

3.65

213.03

1,598.93

homo/Auricularia_subglabra

24,298

7,050.30

1,019.26

4.09

249.44

1,954.16

homo/Dichomitus_squalens

12,791

5,768.41

824.10

3.94

209.25

1,682.65

homo/Trametes_versicolor

13,424

5,530.86

807.15

3.81

211.58

1,678.08

homo/Coprinus_cinereus

11,978

5,943.94

809.24

3.87

209.33

1,791.69

trans.orf/RNAseq

5,981

3,240.91

1,428.35

8.00

329.64

86.35

BUSCO

1,313

3,314.44

2,000.22

10.20

196.06

142.82

MAKER

12,152

2,637.78

1,395.70

6.79

310.75

90.88

HiCESAP

13,803

2,412.73

1,351.37

6.41

300.84

89.62

  1. Line 99: which experiment was replicated? Does it mean 3 different fruiting bodies?

          Response 3.

          Thank you. It means three cDNA libraries were constructed per fruiting body. We have revised this sentence line 103-105 “cDNA library construction and paired-end sequencing was performed with three technical replicates performed per sample using the Illumina HiSeq X Ten platform (San Diego, CA, USA) at the Tianjin Novogene Bioinformatics Technology Co., Ltd.”.

  1. Line 251: be careful, the footnote moved partly vertically.

         Response 4.

         Thank you. We have revised it.

  1. Line 281: PAV is defined in abstract, but it should be also defined here: presence/absence variation

         Response 5.

         Thank you. As suggested, we have defined PAV.

          Line 278. “presence/absence variation (PAV)”.

  1. Line 293: Group VI, maybe you referred to group II.

         Response 6.

         Sorry for this mistake. We have revised “Group VI” to “Group II-C”.

  1. Line 308: Change mian to main.

         Response 7.

         Sorry for this mistake. We have revised.

  1. Line 353: 67.56% could be deleted here as it is in the previous sentence.

         Response 8.

         Sorry for this mistake. We have deleted.
